# miR-252 targeting temperature receptor *CcTRPM* to mediate the transition from summer-form to winter-form of *Cacopsylla chinensis*

Songdou Zhang, Jianying Li, Dongyue Zhang, Zhixian Zhang, Shili Meng, Zhen Li, Xiaoxia Liu*

Department of Entomology and MOA Key Lab of Pest Monitoring and Green Management, College of Plant Protection, China Agricultural University, Beijing, China

**\*For correspondence:**
liuxiaoxia611@cau.edu.cn

**Competing interest:** The authors declare that no competing interests exist.

**Abstract** Temperature determines the geographical distribution of organisms and affects the outbreak and damage of pests. Insects seasonal polyphenism is a successful strategy adopted by some species to adapt the changeable external environment. *Cacopsylla chinensis* (Yang & Li) showed two seasonal morphotypes, summer-form and winter-form, with significant differences in morphological characteristics. Low temperature is the key environmental factor to induce its transition from summer-form to winter-form. However, the detailed molecular mechanism remains unknown. Here, we firstly confirmed that low temperature of 10 °C induced the transition from summer-form to winter-form by affecting the cuticle thickness and chitin content. Subsequently, we demonstrated that *CcTRPM* functions as a temperature receptor to regulate this transition. In addition, miR-252 was identified to mediate the expression of *CcTRPM* to involve in this morphological transition. Finally, we found *CcTre1* and *CcCHS1*, two rate-limiting enzymes of insect chitin biosyntheis, act as the critical down-stream signal of *CcTRPM* in mediating this behavioral transition. Taken together, our results revealed that a signal transduction cascade mediates the seasonal polyphenism in *C. chinensis*. These findings not only lay a solid foundation for fully clarifying the ecological adaptation mechanism of *C. chinensis* outbreak, but also broaden our understanding about insect polymorphism.

## eLife assessment

This is a **valuable** study of the molecular basis of summer-to-winter transition in the pear psyllid pest, *Cacopsylla chinensis* (hemiptera). The molecular and organismal experiments using current methodologies to evaluate the cold responsiveness of the target proteins are mostly **convincing**, but the structural and phylogenetic analyses remain inconclusive. The results of this study will be of interest to entomologists.

## Introduction

Currently, global climate change is the most important environmental problem facing with mankind, and also one of the most complex challenges in the 21st century. Insects, as the largest animal population on the earth, adapt to the complex and changeable external environment by displaying polyphenism, thus achieving the purpose of survival and reproduction (*Simpson et al., 2011*). Insect polyphenism is the phenomenon allows them within a given genotype to produce two or more distinct phenotypes

under the challenges posed by environmental variability via beneficial shifts in morphology, behavior, or physiology (*Cridge et al., 2015*). Polyphenism is commonly observed in many insect species, for examples: the solitary and gregarious phases of *Locusta migratoria* (*Ma et al., 2011*), the short-winged and long-winged morph of *Nilaparvata lugens* (*Xu et al., 2015*), the wingless and winged aphid of *Acyrthosiphon pisum* (*Vellichirammal et al., 2017*), and the catkin and twig morphs in caterpillars of the moth *Nemoria arizonaria* (*Simpson et al., 2011*). With the global climate problems are increasingly prominent, the study of insect polyphenism has become a new hotspot in insect ecology, epigenetics and developmental biology.

Pear psylla is a serious pest of pear trees worldwide, including *Cacopsylla chinensis* (Yang & Li), *Psylla Betulaefoliae*, *Psylla Pyri*, and so on *Butt and Stuart, 1986*; *Wei et al., 2020*. *C. chinensis*, belongs to Hemiptera: Psyllidae, which exclusively feeds on pear trees, harms the young shoots and leaves with adult and nymph, and secretes a large amount of honey dew to breed coal pollution bacteria of *Gloeodes pomigena* (*Wei et al., 2020*). It is also an important vector of the destructive disease of pear trees, *Erwinia amovora*, and currently one of the key pests in the major pear production areas in many East Asian countries, including China and Japan (*Hildebrand et al., 2000*; *Ge et al., 2019*). In most pear production areas of China, the summer-form of *C. chinensis* are usually present from April to September, whereas winter-form of *C. chinensis* are present during the rest of the year (*Ge et al., 2019*; *Wei et al., 2020*). The summer-form has lighter body color and stronger activity ability, and causes more serious damage compare with winter-form. For better adapt to the harsh environmental conditions, the eggs of summer-form develop to the winter-form before the pear fallen leaves. The winter-form has brown to dark brown body color and larger body size, but has the weaker activity ability and stronger resistance ability compare with summer-form. With the gradual increase of temperature in early spring, the winter-form lays eggs at the tender buds of pear trees and then develop into summer-form (*Ge et al., 2019*; *Tougeron et al., 2021*). Hence, the transition between summer-form and winter-form is a significant ecological strategy for *C. chinensis* to avoid adverse environmental conditions and then burst into disaster, which has prominence biological and ecological significance. The seasonal polyphenism of summer-form and winter-form in *C. chinensis* has been recognized for many years and could be induced by low temperature and short day (*Oldfield, 1970*; *Tougeron et al., 2021*), but until recently no molecular mechanism had been advanced.

Temperature is a significant factor limiting the geographic ranges of species, which determines the distribution and diffusion area of species. As a type of small poikilothermic animal, insect is sensitive to temperature changes and temperature stress response is one of the most conservative mechanism to resist extreme environment (*Teets et al., 2023*). Different environmental conditions affect the polyphenism of many kinds of insects, for examples, wing dimorphism of Hemiptera *Schizaphis graminum* (*An et al., 2012*), diapause of Lepidoptera *Bombyx mori* (*Tsuchiya et al., 2021*) and Coleoptera *Colaphellus bowringi* (*Guo et al., 2021*). Under the low temperature conditions, insects can improve cold tolerance by regulating their internal energy metabolism and external exoskeleton system. Such as: darken their cuticles with melanin (*Fedorka et al., 2013*), thickened wax layer or chorion (*Kreß et al., 2016*), and store more lipids to increase energy supply and increase the accumulation of anti-freeze protective substances (*Terblanche et al., 2011*). However, studies about how insect temperature receptor response to temperature change and then activates the downstream signal molecules to regulate the polyphenism have been very limited.

Transient receptor potential (TRP) channel family gene was firstly found in *Drosophila melanogaster* (*Minke et al., 1975*). The TRP channel genes involved in temperature sensing are called 'thermo-sensitive TRP' or 'temperature receptor' (*Lee et al., 2005*). In mammals, there are six thermo-sensitive TRP receptor genes, of which TRPV1-4 mainly feel moderate high temperature (*Venkatachalam and Montell, 2007*) and TRPM8 can be activated by low temperature (8°C–28°C) or cooling substances, such as menthol and borneol (*Peier et al., 2002*). The activation mode is mainly due to the depolarization of cell membrane caused by the influx of $Ca^{2+}$ (*McKemy et al., 2002*). Presently, researches on the function of temperature receptor gene TRP were mainly focused on some model insects, but very few on the agricultural pests. For example, *D. melanogaster TRPM* is highly expressed in dendritic sensory neurons and can sense cold stimulation (*Turner et al., 2016*). In *B. mori*, high temperature receptor *BmTRPA1* regulates the diapause of offspring eggs by acting on the neuroendocrine hormone signal of GABAergic-GnRH/Crz-DH (*Sato et al., 2014*; *Tsuchiya et al., 2021*). However, studies on agricultural insects, such as *Apolygus lucorum* (*Fu et al., 2016*), *N. lugens* (*Wang et al., 2021b*), *Bemisia*

*tabaci* (*Dai et al., 2018*), and *Tuta absoluta* (*Wang et al., 2021a*), also only focused on the identification of the related temperature receptor gene TRPs and its correlation with temperature, no detailed molecular mechanism were reported. In agricultural insects, members of TRPA subfamily are mainly responsible for answering the high-temperature stimulation (*Tracey et al., 2003*; *Neely et al., 2011*), while members of TRPM subfamily are mainly involved in low-temperature stimulation as temperature receptors (*Rosenzweig et al., 2008*; *Turner et al., 2016*). Temperature significantly affected the annual generations, adult survival rate, hibernation period, and egg laying period of *C. chinensis* (*Butt and Stuart, 1986*). Our preliminary research results found that low temperature is the key factor to induce the transition from summer-form to winter-form in *C. chinensis*. Therefore, we speculated that *TRPM*, functions as a temperature receptor gene, may participate in the transition from summer-form to winter-form in response to low temperature, but the specific molecular mechanism needs to be further explored.

MicroRNAs (miRNAs), a class of non-coding RNA molecules, were encoded by endogenous genes in a length of about 18–25 nt. They mainly regulate many life processes, including cell differentiation, organ development, cell apoptosis, biological rhythm and metamorphosis development in animals and plants at the post-transcriptional level by inhibiting the translation process of target genes or degrading the mRNA of target genes (*Asgari, 2013*). In recent years, more and more studies reported that miRNAs are involved in regulating the insect polymorphism, such as the caste differentiation of honeybees, the phenotypic plasticity of *L. migratoria* and the wing pleomorphism of *N. lugens* (*Liu et al., 2012*; *Yang et al., 2014*; *Ye et al., 2019*). In addition, miRNAs also play important roles in insect adaptation to temperature stress. For examples, miR-7 and miR-8 involved in the process of temperature adaptation of *D. melanogaster* (*Kennell et al., 2012*). MiRNAs were also found in responding to low temperature stress in rice water weevil (*Lissorhoptrus oryzophilus*) and gall midge (*Epiblema Scudderiana*) (*Yang et al., 2017b*; *Lyons et al., 2015*). There were also reported that miR-204 targeted *TRPM3* to regulate mammalian eye development (*Shiels, 2020*), and miR-135a targeted *TRPC1* to participate in podocyte injury (*Yang et al., 2017a*). Here, we screened and validated miR-252, a miRNA whose expression was down-regulated under low temperature, negatively targeted *CcTRPM* to modulate the transition from summer-form to winter-form by acting on the cuticle thickness and cuticle chitin content. Chitin biosynthesis signal was also proved to involve in the transition from summer-form to winter-summer in response to *CcTRPM* and miR-252. Our work not only promotes the comprehensive understanding of the adaptation mechanism for *C. chinensis* outbreak at the physiological and ecological level as well as the insect polymorphism, but also helpful in identifying potential target genes in pest control through blocking temperature adaptation of pests.

## Results
### Low temperature induced the transition from summer-form to winter-form in *C. chinensis*

*C. chinensis* was displayed as seasonal polyphenism along with seasonal changes: the summer-form and the winter-form. The significant differences in morphological characteristics are as following: (1) Eggs are long oval shape with a thin tip and a blunt end. Eggs of summer-form are milky white at the initial stage and gradually become yellow white at the later stage (*Figure 1A*). Eggs of winter-form are mostly yellowish brown or orange (*Figure 1A'*). (2) The nymph has five instars and its body size is gradually increasing with age. The body of the newly hatched nymph is oval and the compound eyes are red. The 1st instar nymph of summer-form is light yellow and transparent (*Figure 1B*). The 3rd and 5th instar nymphs of summer-form are yellow-green or blue-yellow, and the wing buds are oblong, protruding and sticking to both sides of the body (*Figure 1C and D*). The 1st instar nymph of winter-form has many black stripes on the chest back plate, and the end of the abdomen and back are black (*Figure 1B'*). The 3rd and 5th instar nymphs of winter-form are overall black brown, with dark brown stripe on the left and right sides of each section on the chest and back of the abdomen, and the end of the abdomen and back becomes a large piece with black brown color (*Figure 1C' , and D'*). (3) The antennae of adults are filamentous with 10 nodes, and the front wings are translucent and oblong. The body color of summer-form adult is green or yellow-green with yellow-brown stripes on the shield of the chest and back plate, and no stripes on the wings (*Figure 1E*). The body color of winter-form adult

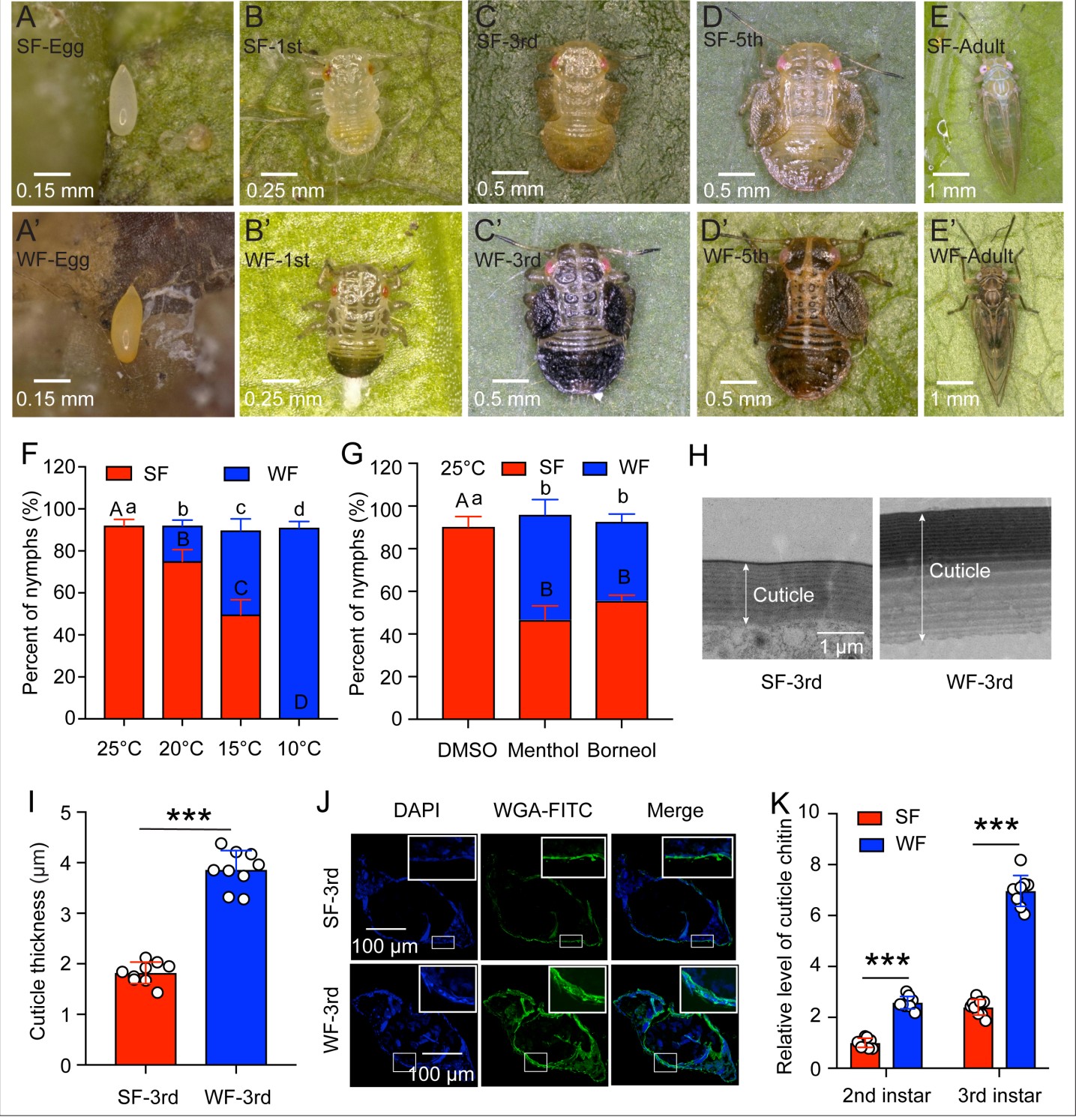

**Figure 1.** Low temperature induced the transition from summer form (SF) to winter form (WF) in *C. chinensis*. (**A-E and A'-E'**) The morphology of egg, 1st, 3rd, and 5th instar nymph, 5 days adult for SF and WF of *C. chinensis*, respectively. The scale bars were labeled in the lower left of each figure. (**F**) Percent of SF and WF under different temperatures (25 °C, 20 °C, 15 °C, and 10 °C) when induction of newly-hatched (within 12 hr) 1st instar nymph of *C. chinensis*. (**G**) Effect of two chemical cooling agents (menthol and borneol) on the percent of SF and WF when induction of newly-hatched (within 12 hr) 1st instar nymph of *C. chinensis* under 25 °C. (**H**) Ultrastructure comparison of the nymph cuticle for SF 3rd instar and WF 3rd instar of *C. chinensis* using transmission electron microscopy. Scale bar is 1 μm. The two-way arrow indicated the cuticle thickness. (**I**) Comparison of the nymph cuticle thickness for SF 3rd instar and WF 3rd instar of *C. chinensis*. (**J**) Chitin staining of nymph cuticle with WGA-FITC in SF 3rd instar and WF 3rd instar of

*Figure 1 continued on next page*

*Figure 1 continued*

*C. chinensis*. DAPI: the cell nuclei were stained with DAPI and visualized in blue. WAG-FITC: the cuticle chitin was labeled with FITC and visualized in green. Merge: merged imaging of DAPI, and WAG-FITC signals. Scale bar is 100 µm. (**K**) Determination of cuticle chitin content in 2nd and 3rd instar nymph of SF and WF. The data of nymph percent were shown as the mean ± SD with three independent biological replications of at least 30 nymphs for each biological replication. Different letters above the bars indicated statistically significant differences (*P*<0.05), as determined by ANOVA followed with a Turkey's HSD multiple comparison test in SPSS 20.0 software. The results of cuticle thickness and cuticle chitin content were presented as mean ± SD of three independent biological samples with three technical replications for each biological replication. Statistically significant differences were determined with the pair-wise Student's *t*-test, and significance levels were denoted by \*\*\* (p<0.001).

is brown or dark brown, with dark brown stripes on the whole body, and clearly brown stripes on the rear edges of the front wings (***Figure 1E'***).

To clarify the effect of temperatures on the transition from summer-form to winter-form, the newly-hatched 1st instar summer-form nymphs were treated with four different temperatures (25 °C, 20 °C, 15 °C, and 10 °C). The results indicated that all the survived treated nymphs develop to the 3rd instar summer-form nymphs at 25 °C, approximately 17% and 40% of survived treated nymphs develop to the 3rd instar winter-form nymphs at 20°C and 15°C, but all the survived treated nymphs develop to the 3rd instar winter-form nymphs at 10 °C (***Figure 1F***). It meant that low temperature (especially 10 °C) was the key environment factor to induce the transition from summer-form to winter-form and 10 °C was selected for the following study. In addition, two cooling agents, menthol and borneol, were also used to mimic the low temperature effect of 10 °C. We observed that about 49% and 37% of the survived treated nymphs develop to the 3rd instar winter-form nymphs after menthol and borneol treatment at 25 °C, respectively (***Figure 1G***).

To further explore the differences between summer-form and winter-form nymphs, cuticle thicknesses and cuticle chitin content were determined. The ultrastructure results of transmission electron microscopy assays showed that cuticle thicknesses of 3rd instar winter-form (about 3.86 µm) were obviously higher than 3rd instar summer-form (about 1.82 µm) (***Figure 1H and I***). We also found that winter-form nymphs exhibited higher chitin level in cuticle compared to the summer-form nymphs by using the WGA-FITC staining and chitin content determination (***Figure 1J and K***). Taken together, these findings demonstrated that low temperature of 10 °C induced the transition of summer-form to winter-form by affecting the cuticle thicknesses and cuticle chitin content.

## *CcTRPM* is a conventional cold-sensing TRP channel activated by 10°C in *C. chinensis*

A cDNA fragment, which was putatively encoded a cold-sensing TRPM channel gene, was identified as *CcTRPM* from *C. chinensis*. The full-length sequence of *CcTRPM* was amplified and verified by RT-PCR with specific primers (***Supplementary file 1a***) and sequencing. Sequence analysis displayed that the ORF of *CcTRPM* was 4566 bp with a 455 bp 3'UTR and encoded a deduced polypeptide of 1521 amino acids with six transmembrane domains. Multiple alignment analysis of the amino acid sequence indicated that the transmembrane domains of *CcTRPM* shared very high amino acid identity with TRPM sequences from other five selected insect species (***Figure 2—figure supplement 1A***). As shown in ***Figure 2—figure supplement 1B***, phylogenetic analysis revealed that insects TRPM and mammals TRPM belong to different branches in evolution. *CcTRPM* was most closely related to the *DcTRPM* homologue (*Diaphorina citri*, XP_017299512.2) in evolutionary relationship, and both of them belong to important Hemiptera pests of fruit trees. The potential tertiary protein structure of *CcTRPM* was constructed with the online server Phyre[2] and modified with PyMOL-v1.3r1 software, and the conserved ankyrin repeats (ANK) and six transmembrane domains were identified (***Figure 2A***).

The results of temperature treatment indicated that the mRNA expression of *CcTRPM* was obviously induced by 10 °C treatment at 1, 2, 3, 6, and 10 days compared with 25 °C treatment by qRT-PCR (***Figure 2B***, ***Figure 2—figure supplement 2F***). It is well known that the TRPM subfamily channels in vertebrates are activated by a variety of chemical compounds, including menthol (***Peier et al., 2002***). We therefore attempted to identify chemical compounds that either activate or inhibit *CcTRPM*. qRT-PCR results revealed that menthol treatment with all three concentrations significantly increased the mRNA expression of *CcTRPM* and antagonist (HY-112430, MedChemExpress, Shanghai, China) treatment with all three concentrations markedly reduced *CcTRPM* transcription (***Figure 2—figure supplement 2G–H***). To further confirm *CcTRPM* is a cold-sensing TRP channel gene, fluorescence

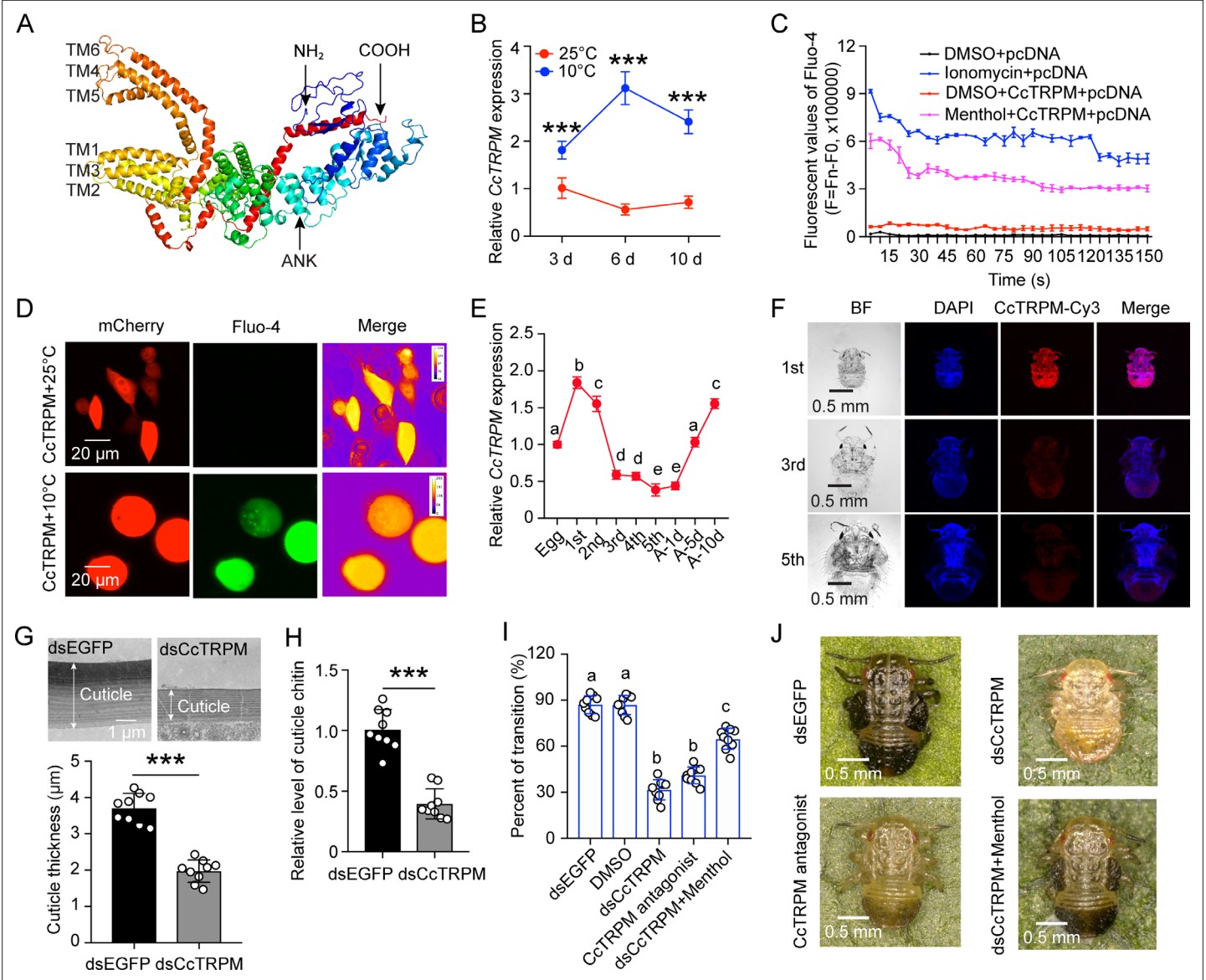

**Figure 2.** Temperature receptor *CcTRPM* modulated the transition from SF to WF of *C. chinensis* in response to low temperature. (**A**) The predicted protein tertiary structure of *CcTRPM*. The conserved ankyrin repeat (ANK) domain was indicated in the N-terminal. The conserved six transmembrane domain of ion channels structure were shown as TM1-TM6. (**B**) The mRNA expression of *CcTRPM* in response to different temperatures of 25°C and 10°C by qRT-PCR. (**C**) Fluorescence detection of Fluo-4 AM after heterologous expression of *CcTRPM* in mammalian HEK293T cells in response to menthol treatment. The recombinant plasmid was generated by inserting the full ORF sequence of *CcTRPM* into pcDNA3.1(+)-mCherry plasmid. DMSO treatment and ionomycin treatment were used as negative control and positive control, separately. (**D**) Representative images of Ca$^{2+}$ imaging after heterologous expression of *CcTRPM* in mammalian HEK293T cells in response to different temperature treatment. "CcTRPM +10 °C" means the recombinant plasmid of *CcTRPM* with pcDNA3.1(+)-mCherry was treated with 10 °C. "CcTRPM +25 °C" denotes the pcDNA3.1(+)-mCherry plasmid with *CcTRPM* was treated with 25 °C. mCherry and Fluo-4 signal are shown in red and green, respectively. Scale bar is 20 µm. (**E**) The developmental expression pattern of *CcTRPM* for SF at mRNA level using qRT-PCR. 1st, 2nd, 3rd, 4th, and 5th are the nymphs at the first, second, third, fourth and fifth instar, respectively. A-1d, A-5d, and A-10d are the adults at 1 day, 5 days, and 10 days, separately. (**F**) Representative confocal images of *CcTRPM* in different developmental stages of SF using FISH. Scale bar is 0.5 mm. BF: the bright field. DAPI: the cell nuclei were stained with DAPI and visualized in blue. CcTRPM-Cy3: CcTRPM signal was labeled with Cy3 and visualized in red. Merge: merged imaging of BF, DAPI, and CcTRPM-Cy3 signals. (**G–H**) Comparison of the nymph cuticle ultrastructure, cuticle thickness, and cuticle chitin content of SF 1st instar treated with dsCcTRPM and dsEGFP at 15 days. (**I**) The transition percent of SF 1st instar nymphs treated with dsEGFP, DMSO, dsCcTRPM, CcTRPM antagonist and dsCcTRPM +menthol at 15 days under 10 °C. For RNAi experiments, summer-form 1st instar nymphs were fed with dsEGFP (500 ng/µL) or dsCcTRPM (500 ng/µL). To mimic RNAi effect, summer-form 1st instar nymphs were fed with 0.1% DMSO or CcTRPM antagonist (20 ng/µL). For the rescue experiment, summer-form 1st instar nymphs were fed with the mixture of dsCcTRPM (500 ng/µL) and menthol (1 mg/mL). Then, counted the number of summer-form and winter-form individuals and calculated the transition percent. (**J**) The phenotypes of SF 1st instar nymphs treated with dsEGFP, dsCcTRPM, CcTRPM antagonist and

*Figure 2 continued on next page*

*Figure 2 continued*

dsCcTRPM +menthol at 15 days under 10 °C. The data in 2B and 2E are shown as the mean ± SD with three independent biological replications of at least 30 nymphs for each biological replication. Scale bar is 0.5 mm. Data in 2 G and 2 H are presented as mean ± SD with three biological replications of three technical replicates for each biological replication. Data in 2I are presented as mean ± SD with nine biological replications. Statistically significant differences were determined with the pair-wise Student's *t*-test, and significance levels were denoted by *** (p<0.001). Different letters above the bars indicated statistically significant differences (p<0.05), as determined by ANOVA followed with a Turkey's HSD multiple comparison test in SPSS 20.0 software.

The online version of this article includes the following figure supplement(s) for figure 2:

**Figure supplement 1.** Sequence characterization of TRPM from different insect species.

**Figure supplement 2.** Effect of menthol and *CcTRPM* antagonist treatment on the mRNA expression of *CcTRPM* under 10 °C condition.

**Figure supplement 3.** RNAi efficiency of *CcTRPM* and effect of *CcTRPM* knockdown on the phenotype at 25 °C.

detection of Fluo-4 AM and $Ca^{2+}$ imaging was performed to determine whether CcTRPM-expressing cells increased intracellular $Ca^{2+}$ concentration upon 10 °C treatment and menthol treatment. The results of fluorescence detection of Fluo-4 AM indicated that there was no change of $Ca^{2+}$ concentration in CcTRPM-expressing cells was treated with DMSO, but a clear increase of $Ca^{2+}$ concentration was observed when treated with menthol which was similar with the trend of pcDNA-expressing cells was treated with the positive control of ionomycin (*Figure 2C*). The effect of 10 °C and menthol on *CcTRPM* was further tested using $Ca^{2+}$ imaging and showed that both 10 °C and menthol treatment obviously elevated intracellular $Ca^{2+}$ levels in CcTRPM-expressing cells (*Figure 2D*, *Figure 2—figure supplement 2I*). We therefore concluded that *CcTRPM* is a cold-sensitive ion channel activated by 10 °C low temperature and menthol.

## *CcTRPM* modulated the transition from summer-form to winter-form of *C. chinensis* in response to low temperature

Temporal expression profiles by qRT-PCR revealed that *CcTRPM* mRNA expressed in all determined stages, with the highest expression level in the 1st instar nymph, and showed a decreased trend from 2nd instar nymph to adult 1 day (*Figure 2E*). The FISH results also indicated that *CcTRPM* mRNA has a greatest expression level in the 1st instar nymph and relative lower expression in the 3rd and 5th instar nymph (*Figure 2F*).

To study whether *CcTRPM* modulated the transition from summer-form to winter-form of *C. chinensis*, the newly-hatched 1st instar nymphs of summer-form were fed with dsCcTRPM or dsEGFP. For the RNAi efficiency under 10 °C condition, feeding of dsCcTRPM reduced the transcription level of *CcTRPM* by about 61% and 62% compare to dsEGFP feeding at 3 days and 6 days by qRT-PCR (*Figure 2—figure supplement 3A*), and decreased the CcTRPM-Cy3 signal at 6 days by FISH (*Figure 2—figure supplement 3B*). Both results of cuticle ultrastructure and cuticle chitin content determination showed that cuticle thicknesses (about 1.97 μm) and cuticle chitin content of dsCcTRPM-treated nymphs were markedly lower than dsEGFP-treated nymphs (about 3.71 μm) under 10 °C condition (*Figure 2G and H*). Additionally, dsCcTRPM feeding and *CcTRPM* antagonist treatment both significantly decreased the transition percent from summer-form to winter-form, but menthol feeding only partially rescued the transition percent and morphological phenotype under dsCcTRPM feeding (*Figure 2I and J*). To exclude the effect of *CcTRPM* itself on the transition percent and nymph development, we also performed the RNAi assays under 25 °C condition. After successfully interfered the mRNA expression of *CcTRPM*, the nymph development and morphological phenotype were not affected compared to the dsEGFP treatment (*Figure 2—figure supplement 3C-D*). Together, these findings indicated that *CcTRPM* functions as an essential molecular signal to modulate the transition from summer-form to winter-form of *C. chinensis* in response to low temperature.

## Confirming of miR-252 targeted *CcTRPM*

To explore the regulation mechanism of *CcTRPM* at the post-transcriptional level, two software programs of miRanda and Targetscan were used to predict the potential miRNAs. There are four miRNAs (miR-252, miR-5728, PC-3p-173924, and PC-3p-137568) were predicted to have the corresponding target-binding sites with the 3'UTR of *CcTRPM* (*Figure 3A* and *Figure 3—figure supplement 1A*). miR-252 showed the opposite expression patterns with *CcTRPM* after 25°C and 10°C treatments (*Figures 2B*

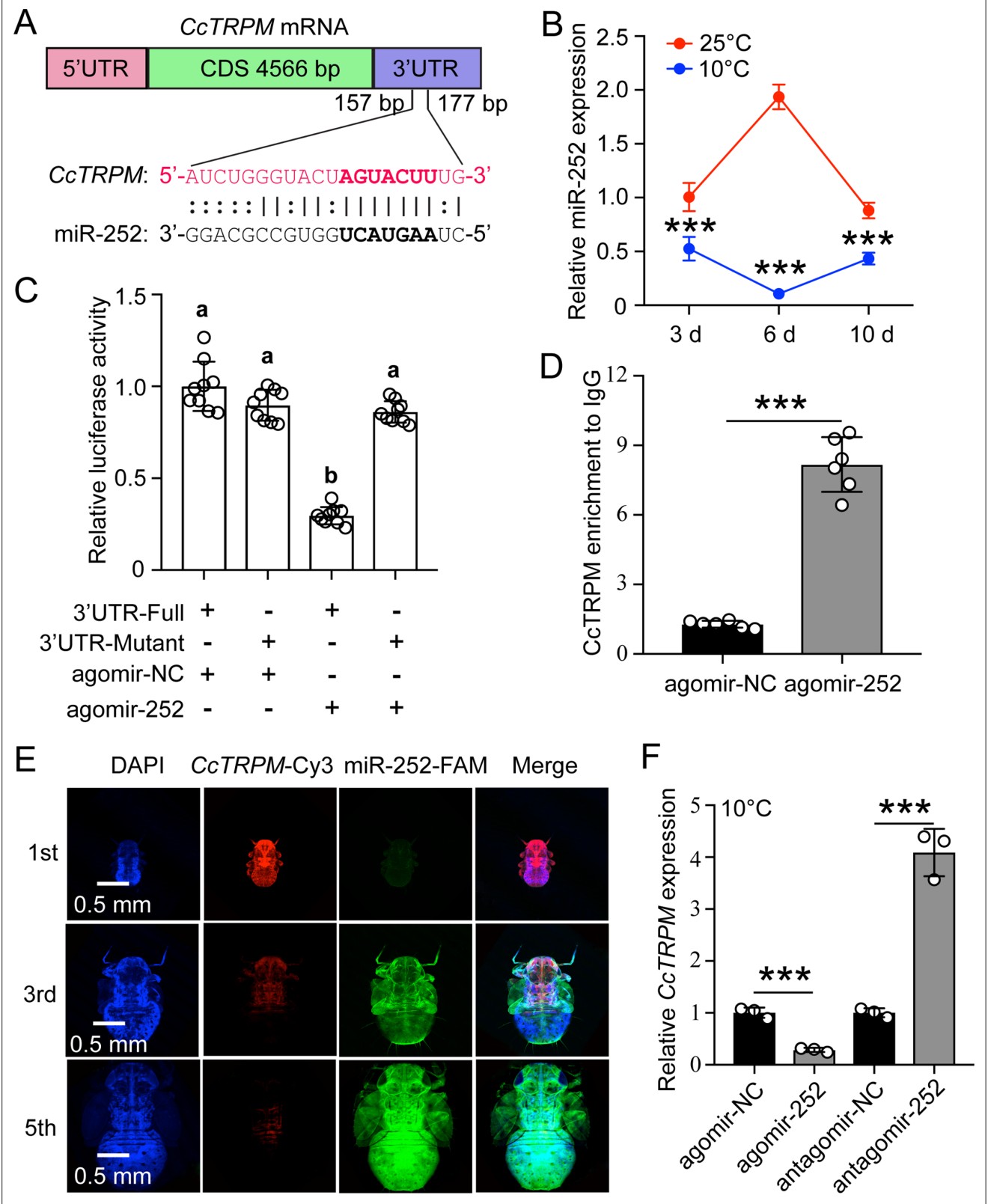

**Figure 3.** Confirming of miR-252 targeted *CcTRPM*. (**A**) The putative miR-252 binding sites in *CcTRPM* 3'UTR were predicted by miRanda and Targetscan. (**B**) The expression profiles of miR-252 in response to different temperatures of 25°C and 10°C by qRT-PCR. (**C**) In vitro validation of the target interactions between miR-252 and *CcTRPM* by dual luciferase reporter assays. (**D**) In vivo demonstration of miR-252 targeting *CcTRPM* by RNA-binding protein immunoprecipitation (RIP) assay. (**E**) Representative confocal images of miR-252 and *CcTRPM* in 1st, 3rd and 5th instar of SF. Scale bar is

*Figure 3 continued on next page*

*Figure 3 continued*

0.5 mm. The signals of DAPI and *CcTRPM*-Cy3 are same as the above describing. miR-252-FAM: the miR-252 signal was labeled with FAM and visualized in green. Merge: merged imaging of co-localization of cell nucleus, *CcTRPM* and miR-252. (F) Effect of miR-252 agomir and antagomir treatment 3 days on the expression of *CcTRPM* at mRNA level. The data in 3B and 3 F are shown as the mean ± SD with three independent biological replications of at least 30 nymphs for each biological replication. Data in 3 C are presented as mean ± SD with nine biological replications. Data in 3D are presented as mean ± SD with two biological replications of three technical replicates for each biological replication. Statistically significant differences were determined with the pair-wise Student's *t*-test, and significance levels were denoted by *** (p<0.001). Different letters above the bars indicated statistically significant differences (p<0.05), as determined by ANOVA followed with a Turkey's HSD multiple comparison test in SPSS 20.0 software.

The online version of this article includes the following figure supplement(s) for figure 3:

**Figure supplement 1.** Validation of the target interactions between *CcTRPM* and other three miRNAs.

**Figure supplement 2.** Transcript of miR-252 was significantly enriched by antibody against Ago1 in agomir-252 treated group compared with agomir-NC group.

*and 3B*). To further determine whether miR-252 specifically targeted *CcTRPM*, the following four experimental approaches were performed: (1) The dual-luciferase reporter assays were carried out by cloning a 419 bp 3'UTR fragment of *CcTRPM* containing the predicted target binding sites of the miRNAs. The relative luciferase activities were obviously decreased when the agomir-252 and CcTRPM-3'UTR-pmirGLO were cotransfected, while the activity recovered to the control when using the mutated 3'UTR sequence without the binding sites of miR-252 (*Figure 3C*). However, the relative luciferase activities were no change when the construct of CcTRPM-3'UTR-pmirGLO was cotransfected with the agomir of other three miRNAs, respectively (*Figure 3—figure supplement 1B*). (2) An RNA immunoprecipitation assay indicated that the transcripts of *CcTRPM* and miR-252 increased 6.35-fold and 9.21-fold from the agomir-252 feeding nymphs of the Ago-1 antibody-mediated RNA complex compared to an IgG control, separately (*Figure 3D* and *Figure 3—figure supplement 2*). (3) FISH results showed that *CcTRPM* and miR-252 were co-expressed in the nymphs and displayed an opposite expression trend along with the increasing instar (*Figure 3E*). (4) qRT-PCR results indicated that agomir-252 feeding significantly inhibited *CcTRPM* mRNA expression and antagomir-252 feeding markedly increased *CcTRPM* mRNA expression compare to their controls, separately (*Figure 3F*). These data together confirmed that miR-252 directly targeted *CcTRPM*.

## miR-252 mediated the transition from summer-form to winter-form in *C. chinensis* by suppressing *CcTRPM*

To test the efficiency of agomir-252, 1st instar nymphs of summer-form were fed with agomir-252 under 10 °C condition. qRT-PCR results showed that miR-252 expression was both significantly increased after agomir-252 treatment at 3 days and 6 days (*Figure 4A*). To further clarify the relationship between *CcTRPM* and miR-252, we determined miR-252 expression after dsCcTRPM treatment under 10 °C condition. Interestingly, miR-252 expression was dramatically inhibited after successful interference of *CcTRPM* at 3 days and 6 days (*Figure 4B*). This result implied that up-regulation of *CcTRPM* expression may have a positive feedback to miR-252 expression in response to 10 °C temperature.

To deeply decipher the role of miR-252 in the transition from summer-form to winter-form, 1st instar nymphs of summer-form were fed with agomir-NC or agomir-252 under 10 °C condition. Results of cuticle ultrastructure indicated that cuticle thickness of agomir-252 treated nymphs (about 1.85 µm) was markedly lower than agomir-NC treated nymphs (about 3.70 µm) (*Figure 4C and D*). Both WAG-FITC staining and chitin content determination revealed that cuticle chitin content of agomir-252 treated nymphs were obviously lower than agomir-NC treated nymphs (*Figure 4E and F*). Moreover, agomir-252 feeding significantly decreased the transition percent from summer-form to winter-form, but antagomir-252 feeding rescued the defect of transition percent and morphological phenotype under dsCcTRPM treatment (*Figure 4G-H*, *Supplementary file 1b*). To further evaluate the effect of agomir-252 under 25 °C condition, we determined *CcTRPM* expression and observed the nymph development after agomir-252 feeding. As expected, *CcTRPM* expression was still significantly decreased after agomir-252 feeding (*Figure 4—figure supplement 1C*), but the nymph development and morphological phenotype were not affected by agomir-252 treatment compared to the agomir-NC treatment (*Figure 4—figure supplement 1D*). Above all, these results displayed that

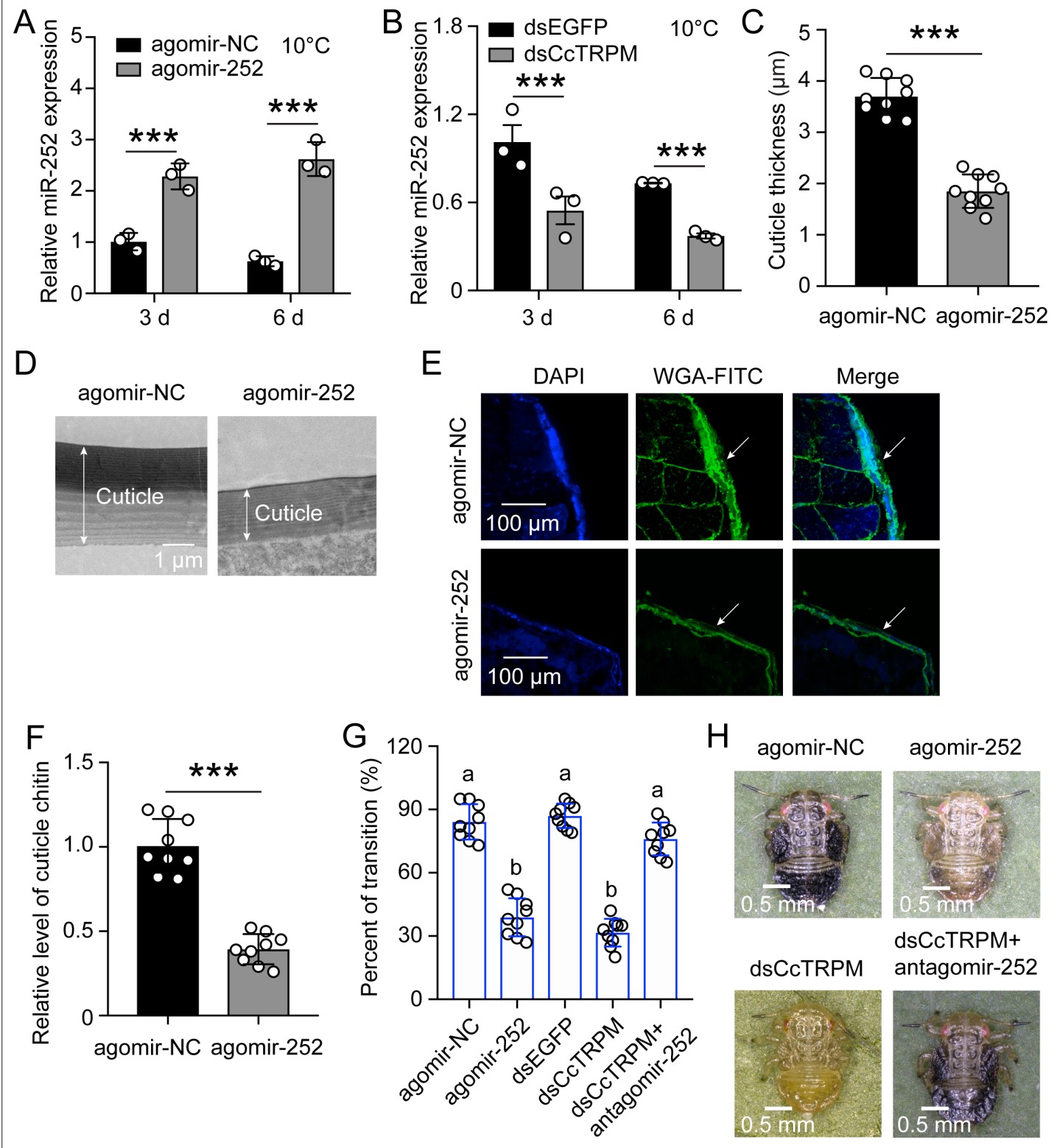

**Figure 4.** miR-252 mediated the transition from SF to WF in *C. chinensis* by suppressing *CcTRPM*. (**A**) Effect of miR-252 agomir treatment on the mRNA expression of miR-252 after 3 days and 6 days under 10 °C. agomir-NC treatment was used as the control. (**B**) The mRNA expression of miR-252 after dsCcTRPM treatment 3 days and 6 days compare with the dsEGFP treatment under 10 °C. (**C–F**) Comparison of the nymph cuticle thickness, cuticle ultrastructure, cuticle chitin staining with WGA-FITC, and chitin content of SF 1st instar after treatment with agomir-NC and agomir-252 at 15 days. Scale bar in (D) is 1μm and in (E) is 100 μm. The two-way arrow indicated the cuticle thickness. The DAPI and WAG-FITC signals were same as the above

*Figure 4 continued on next page*

*Figure 4 continued*

describing. (**G**) The transition percent of SF 1st instar nymphs treated with agomir-NC, agomir-252, dsEGFP, dsCcTRPM, and dsCcTRPM +antagomir-252 at 15 day under 10 °C. (**H**) The phenotypes of SF 1st instar nymphs treated with agomir-NC, agomir-252, dsCcTRPM, and dsCcTRPM +antagomir-252 at 15 day under 10 °C. Scale bar is 0.5 mm. The data in 4A and 4B are shown as the mean ± SD with three independent biological replications of at least 30 nymphs for each biological replication. Data in 4C and 4F are presented as mean ± SD with three biological replications of three technical replications for each biological replication. Data in 4G are presented as mean ± SD with nine biological replications. Statistically significant differences were determined with the pair-wise Student's *t*-test, and significance levels were denoted by \*\*\* (p<0.001). Different letters above the bars indicated statistically significant differences (p<0.05), as determined by ANOVA followed with a Turkey's HSD multiple comparison test in SPSS 20.0 software.

The online version of this article includes the following figure supplement(s) for figure 4:

**Figure supplement 1.** Effect of agomir-252 on the phenotype of SF 1st instar nymphs at 25 °C.

miR-252 mediated the transition from summer-form to winter-form in *C. chinensis* by suppressing *CcTRPM* via affecting the cuticle thickness and cuticle chitin content.

### Chitin biosynthesis signaling involved in the transition from summer-form to winter-summer of *C. chinensis* in response to low temperature and *CcTRPM*

Since cuticle thickness and cuticle chitin content were obviously affected by dsCcTRPM treatment and agomir-252 treatment, we speculated that chitin biosyntheis signaling pathway participated in the transition from summer-form to winter-form. To demonstrate this hypothesis, a total of 14 transcripts corresponding to 8 genes involved in chitin biosyntheis were identified in *C. chinensis*, with several genes were present in two or more paralogous forms (*Figure 5A*). Next, we used qRT-PCR to determine the expression profiles of these 14 chitin biosyntheis transcripts after 25°C and 10°C treatments at 3 days, 6 days, and 10 days. Based on the relative gene expression in the heat map, we found that the expression of 13 transcripts exhibited overall up-regulation under 10 °C condition at 3 days, 6 days, and 10 days compared to those under 25 °C condition, while levels for CcUAP2 were almost unchanged (*Figure 5B*).

Subsequently, we used RNAi to further uncover the impact of chitin biosyntheis genes in the transition from summer-form to winter-form. Compared with dsEGFP feeding control, only knockdown of *CcTre1* and *CcCHS1* obviously decreased the transition percent from summer-form to winter-form, but not by knockdown of the remaining transcripts (*Figure 5C*). To further dissect the roles of *CcTre1* and *CcCHS1*, cuticle thickness and cuticle chitin content were also determined. Compared with dsEGFP feeding control, knockdown of *CcTre1* and *CcCHS1* markedly reduced the cuticle thickness and cuticle chitin content (*Figure 5D–F*, *Figure 5—figure supplement 1C–D,*), which were similar with dsCcTRPM feeding and agomir-252 treatment. To confirm whether *CcTre1* and *CcCHS1* function as the downstream of *CcTRPM* and miR-252, we next determine the expression of *CcTre1* and *CcCHS1* after *CcTRPM* knockdown and agomir-252 feeding. qRT-PCR results indicated that *CcTRPM* knockdown and agomir-252 feeding were both obviously reduced the mRNA expression of *CcTre1* and *CcCHS1* at 3 days and 6 days, respectively (*Figure 5G–J*). These phenotypic and molecular data suggested that chitin biosynthesis signaling involved in the transition from summer-form to winter-summer of *C. chinensis* in response to low temperature and *CcTRPM*.

## Discussion

Polyphenism offers the opportunity for insects to deploy different phenotypes generated from the same genome to best suit the extreme environmental changes (*Simpson et al., 2011*). In recent years, studies on insect polyphenism have mainly focused on the molecular regulation by endocrine hormones, neuropeptides, and neurotransmitters, while the regulation mechanism by external environmental factors remains poorly understood. For examples, ecdysone signaling and insulin/IGF signaling pathway regulated the wing polyphenism of pea aphid (*Vellichirammal et al., 2017*) or planthoppers (*Zhang et al., 2022*), neuropeptide F was essential for shaping locomotor plasticity in locust (*Hou et al., 2017*), and dopamine metabolic pathway modulated the behavioral phase changes of the migratory locust (*Ma et al., 2011*). Temperature is important in defining the distribution of insects, but how it activates the temperature receptor and then regulates polyphenism need further deeply study. Here, we firstly demonstrated that the transition from summer-form to winter-form in *C.

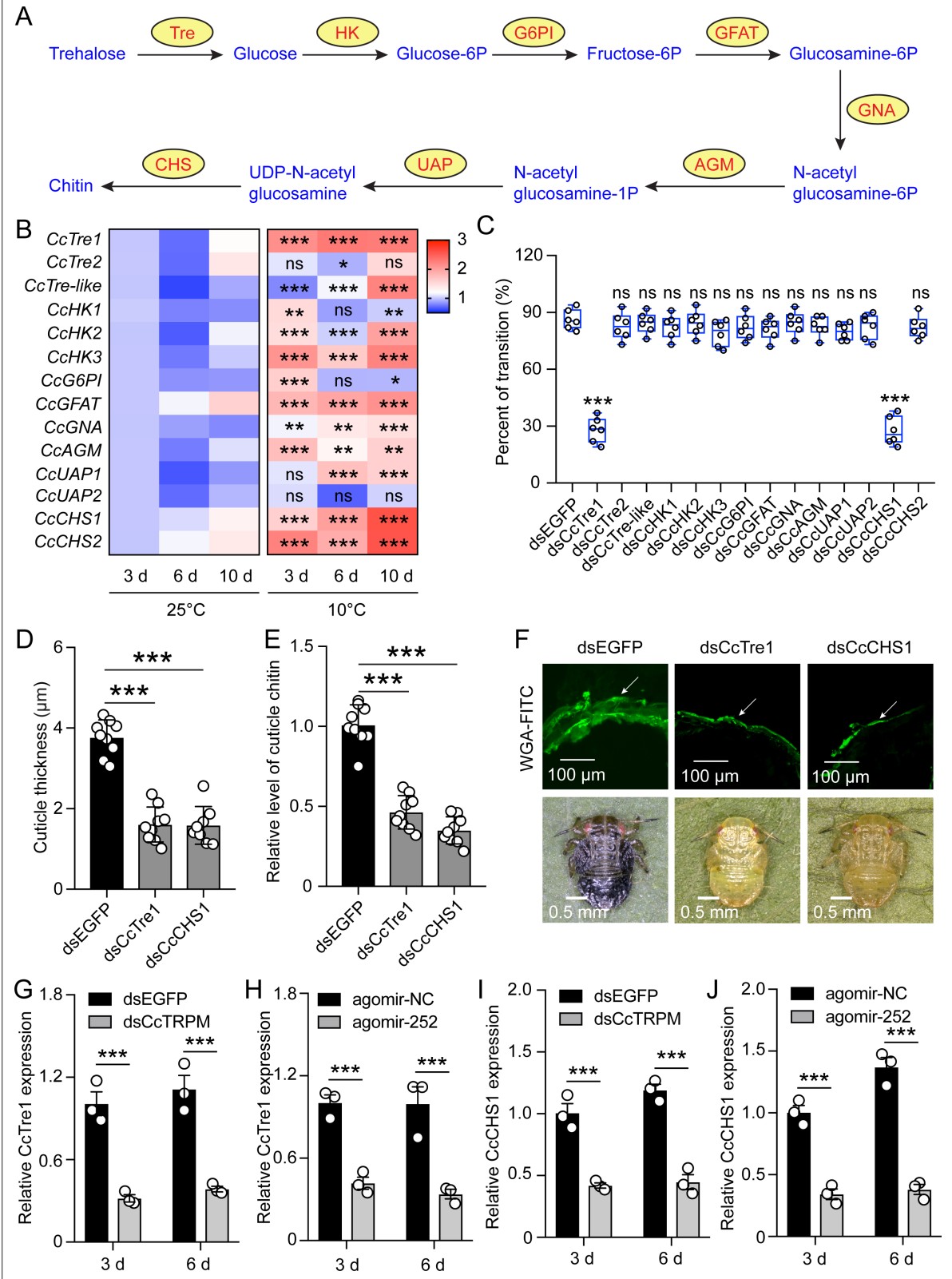

**Figure 5.** Chitin biosynthesis signaling involved in the transition from SF to WF of *C. chinensis* in response to low temperature and *CcTRPM*. (**A**) Diagram of de novo biosynthesis of insect chitin. Tre: trehalase. HK: hexokinase. G6PI: glucose-6-phosphate isomerase. GFAT: fructose-6-phosphate aminotransferase. GNA: glucosamine-6-phosphate acetyltransferase. AGM: N-acetylglucosamine phosphate mutase. UAP: UDP- N-acetylglucosamine pyrophosphorylase. CHS: chitin synthase. (**B**) A heat map was constructed from the expression levels of chitin biosynthesis enzyme transcripts in *C.*

*Figure 5 continued on next page*

*Figure 5 continued*

*chinensis* after 25°C and 10°C treatment at 3 d, 6 d, and 10 d. (**C**) Effect of RNAi knockdown of 14 transcripts covering all the enzymes in the chitin biosynthesis pathway on the transition percent of SF 1st instar nymphs under 10 °C. (**D–F**) Comparison of the nymph cuticle thickness, cuticle chitin content, and cuticle chitin staining with WGA-FITC of SF 1st instar after treatment with dsEGFP, dsCcTre1, and dsCcCHS1 at 15 d. The WAG-FITC signal was same as the above describing. Scale bar in (**F**) is 100 μm and 0.5 mm, respectively. (**G–H**) Effect of dsCcTRPM and agomir-252 treatments on the mRNA expression of *CcTre1* at 3 day and 6 day under 10 °C. (**I–J**) Effect of dsCcTRPM and agomir-252 treatments on the mRNA expression of *CcCHS1* at 3 day and 6 day under 10 °C. The data in (**C**) are shown as the mean ± SD with six independent biological replications of at least 30 nymphs for each biological replication. Data in (**D and E**) are presented as mean ± SD with three biological replications of three technical replications for each biological replication. Data in (**G-J**) are presented as mean ± SD with three biological replications. Statistically significant differences were determined with the pair-wise Student's *t*-test, and significance levels were denoted by * (p<0.05), ** (p<0.01), and *** (p<0.001).

The online version of this article includes the following figure supplement(s) for figure 5:

**Figure supplement 1.** RNAi efficiency of *CcTre1* and *CcCHS1* at 10 °C by qRT-PCR.

*chinensis* was 'temperature-dependent pattern', and low temperature of 10 °C completely induced this transition. Just like people wear more clothes in the winter, we also proved that 10 °C induced the summer-form change to winter-form by increasing the cuticle thickness and cuticle chitin content. Several similar phenotype changes have been reported in other insects: the cricket of *Allonemobius socius* exhibited darker cuticles to improve thermo-regulation in colder environments (*Fedorka et al., 2013*), the mosquito of *Aedes albopictus* increased the thickness of the egg middle serosa cuticle and wax layer as an adaptation strategy for cold acclimation (*Kreß et al., 2016*).

TRPM8, a member of the TRP family, has been reported as the principal sensor of environmental cold in mammals (*Peier et al., 2002*). As a kind of poikilothermic animals, insects also have an evolutionarily conserved TRPM gene which is homologous to mammalian temperature receptor of TRPM8 (*Venkatachalam and Montell, 2007*). In this study, we identified a classical TRPM gene in *C. chinensis* which was designated as *CcTRPM* and functioned as a low temperature receptor by using qRT-PCR and calcium imaging. After knockdown of *CcTRPM* by feeding dsCcTRPM, the treated nymphs displayed thinner cuticle thickness and less cuticle chitin content compared with the control of dsEGFP treatment under 10 °C condition. In addition, *CcTRPM* knockdown by RNAi or *CcTRPM* antagonist feeding both caused obvious defect of the transition percent from summer-form to winter-form, but this defect could be partially rescued by a cooling agent menthol treatment. To exclude the developmental defect or genetic background, we also knocked down *CcTRPM* under 25 °C condition, and not found any death defect and cuticle defect. These results were the first report about *CcTRPM* was essential for insect polyphenism in response to low temperature. Until now, the researches about TRPM on other agricultural insects only focused on the description of its correlation with temperature, but no deeply molecular function was reported (*Fu et al., 2016*; *Wang et al., 2021b*). Therefore, the molecular regulation mechanism of *CcTRPM* in response to low temperature needs further study. TRPV ion channel has been reported to function as a heat-activated receptor in mammals by David Julius (*Caterina et al., 1997*; *Cao et al., 2013*). So, TRPV is supposed to function as a heat-activated receptor in *C. chinensis*, and it is also worthy to explore whether it induces the transition from winter-form to summer-form in the future.

miRNAs played significant roles in insect polyphenism and cold acclimation. For example, miR-34 and miR-9b mediated the wing dimorphism of *N. lugens* (*Ye et al., 2019*) and *A. citricidus* (*Shang et al., 2020*), miR-133 inhibited behavioral aggregation by controlling dopamine synthesis in *L. migratoria* (*Yang et al., 2014*), and miR-31–5 p via asc-C9 to regulate cold acclimation in *Monochamus alternatus* (*Zhang et al., 2020*). Here, we demonstrated that a key regulator, miR-252, targets *CcTRPM* via a negative feedback loop to control the transition from summer-form to winter-form in response to low temperature. Interestingly, RNAi-mediated knockdown of *CcTRPM* also decreased the expression of miR-252. This result implied that up-regulation of *CcTRPM* in response to low temperature may be have a feedback to miR-252, then miR-252 inhibits the expression of *CcTRPM* under low temperature condition to avoid its excessive expression. Upon miR-252 agomir feeding, the treated nymphs showed thinner cuticle thickness and less cuticle chitin content compared with the control of agomir-NC feeding under 10 °C condition. Besides, miR-252 antagomir feeding could rescue the transition percent defect and morphological phenotype defect caused by *CcTRPM* knockdown. These results were the first report about miR-252 functions as the up-stream regulator to control the transition from summer-form to winter-form in response to low temperature via acting on the insect cuticle

thickness and cuticle chitin content. Similar result was reported in *Daphnia pulex*, miR-252 might increase cadmium tolerance by switching cellular growth and cuticle protein pathways to detoxification processes (***Chen et al., 2016***). In *Leptinotarsa decemlineata*, 42 differentially expressed miRNAs were identified following cold exposure including miR-9a-3p, miR-210–3 p, miR-276–5 p and miR-277–3 p (***Morin et al., 2017***). Therefore, the regulatory network of miRNAs is need further investigating in the transition between two phases in *C. chinensis*, especially transboundary miRNAs from host plant.

Insect cuticle serves as a protective barrier and plays significant roles in supporting the body, resisting the adverse environmental conditions, preventing predators from predation, and preventing pathogens from invading (***Evans and Sanson, 2005***; ***Vargas et al., 2014***). Chitin as an extremely important structural component of insects and represents up to 60% of the dry weight in some species (***Richards, 1978***). Many studies have reported that insect cuticles are essential for cold acclimation (***Fedorka et al., 2013***; ***Kreß et al., 2016***). In this study, cuticle thickness and cuticle chitin content were the major differences between summer-form and winter-form. We have demonstrated that *CcTRPM* and miR-252 function as the up-stream signaling to mediate the transition from summer-form to winter-form in response to low temperature via affecting the cuticle thickness and cuticle chitin content. Therefore, it is necessary to study whether chitin biosynthesis pathway acting as the down-stream signaling in this transition. The de novo biosynthesis of chitin has eight enzymatic steps, including 1 Trehalose, 2 enzymes in Glycolysis, 4 enzymes in Hexosamine pathway, and 1 Chitin synthesis (***Doucet and Retnakaran, 2012***). Further RNAi results showed that two rate-limiting enzyme genes, *CcTre1* and *CcCHS1*, play essential roles in the transition from summer-form to winter-form in response to low temperature and affect the cuticle thickness and cuticle chitin content. Glycolysis and hexosamine pathway are two complex cellular metabolic processes within organisms. We supposed that there are two reasons for not all of these steps were required: (1) the function of some enzymes may be replaced or supplemented by other enzymes, for examples, function of hexokinase and glucokinase was similar. (2) The reason for no obviously phenotypic defects might be cause by insufficient interference efficiency of RNAi. So, it's worth to further study the functions of these chitin biosynthesis enzymes by CRISPR-Cas9 in future. In addition, *CcTRPM* knockdown and miR-252 agomir feeding both markedly decreased the mRNA expression of *CcTre1* and *CcCHS1*. These results proved our hypothesis and were the first report that chitin biosynthesis pathway functions as the down-stream signaling to control the transition from summer-form to winter-form in *C. chinensis*.

In summary, this study reported a novel signaling pathway of low temperature-miR-252-*CcTRPM* acting on cuticle chitin biosynthesis genes to mediate the transition from summer-form to winter-form in *C. chinensis*. We proposed a working model to describe this novel mechanism in ***Figure 6***. Under 25 °C condition, the expression of temperature receptor *CcTRPM* was depressed by miR-252, so the 1st instar nymphs of summer-form normally developed into 3rd instar nymphs of summer-form. However, under 10 °C condition, low temperature greatly activated the expression of *CcTRPM*, but miR-252 functioned as a negative feedback regulator to mediate *CcTRPM* expression to avoid its excessive expression in response to low temperature. Then, *CcTRPM* obviously improved the expression of two rate-limiting enzyme genes from cuticle chitin biosynthesis pathway, *CcTre1* and *CcCHS1*, to increase cuticle thickness and raise cuticle chitin content. Lastly, the 1st instar nymphs of summer-form developed to 3rd instar nymphs of winter-form as an ecological adaptation strategy under low temperature stress. Future researches will focus on the following directions: (1) Whether and how transboundary miRNAs and hormones from host plant involve in the transition from summer-form to winter-form in *C. chinensis*. (2) How neuropeptides and endocrine hormones response to low temperature and *CcTRPM* to mediate the transition from summer-form to winter-form in *C. chinensis*.

## Materials and methods
### Insects and host plants

The summer-form and winter-form colonies of *C. chinensis* (Yang & Li) were initiated collected in June and December 2018 from a pear orchard in Daxing district, Beijing, China, respectively. These two populations are now reared in the integrated pest management (IPM) laboratory of China Agricultural University. The summer-form population was raised in growth chambers at 25 ± 1°C under a photoperiod of 12 L: 12D with 65±5% RH (relative humidity). The feeding conditions of winter-form population

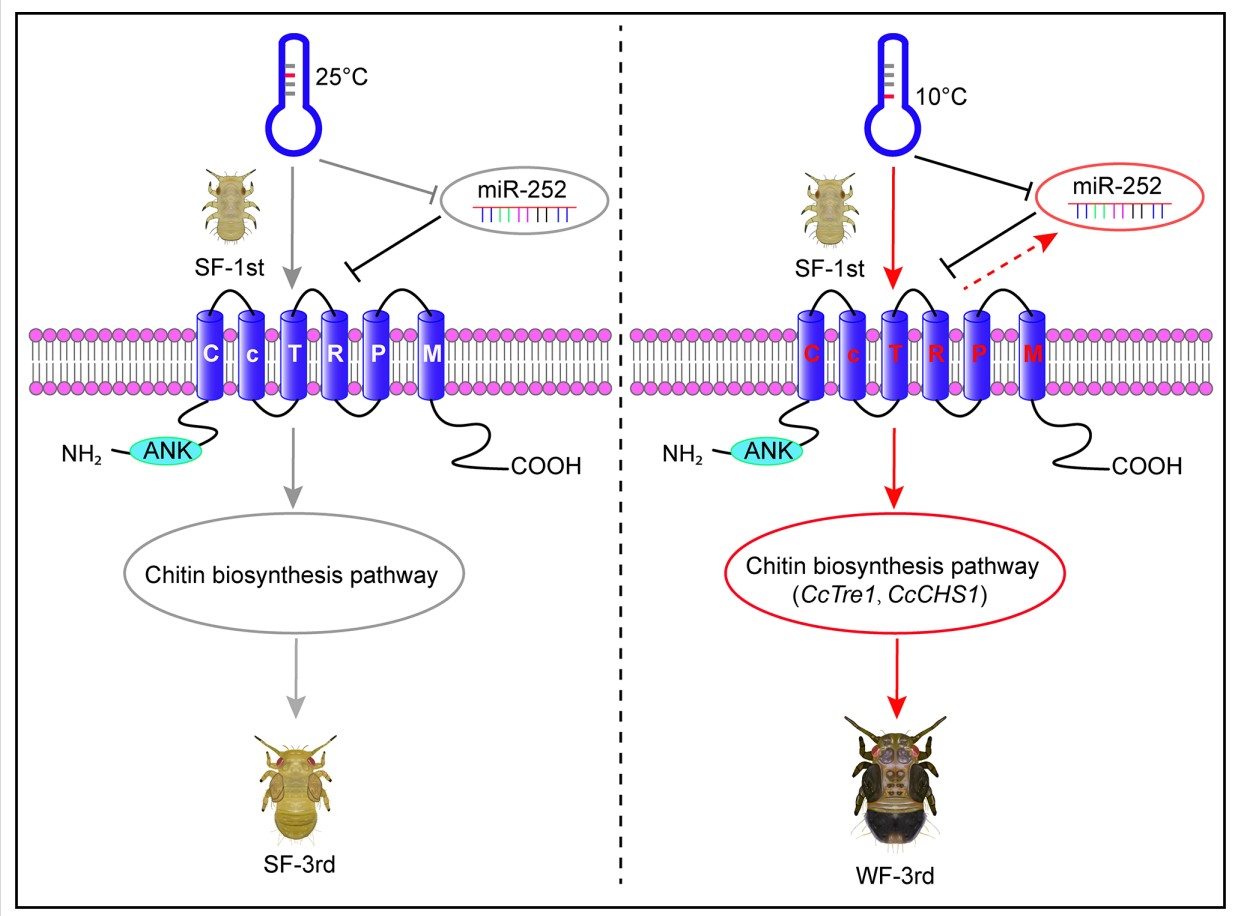

**Figure 6.** A model showing the key roles of miR-252 targeting *CcTRPM* to regulate the transition from SF to WF of *C. chinensis* in response to low temperature. Under 25 °C condition, the temperature receptor *CcTRPM* was inactive as inhibiting by miR-252 in 1st instar nymphs of SF. The 1st instar nymphs of SF normally developed into 3rd instar nymphs of SF. However, under 10 °C condition, low temperature profoundly activated *CcTRPM* expression, miR-252 functioned as a negative regulator to mediate the expression of *CcTRPM* in case of over-expression in response to low temperature. Then, *CcTRPM* significantly increased the activity of chitin biosynthesis pathway, especially *CcTre1* and *CcCHS1*, and led to cuticle chitin content was obvious raise and cuticle thickness became thicker. Finally, the 1st instar nymphs of SF developed to 3rd instar nymphs of WF in order to better adapt to low temperature. Red arrows and black T-bars indicated the activation and inhibition, respectively. Gray arrows and T-bars represented the inactive states of the genes or physiological processes, separately.

were raised at 10 ± 1 ° C under a photoperiod of 12 L: 12D with 25±5% RH. Unless otherwise specified, the photoperiod of all the following different temperature treatments is 12 L: 12D.

Host plants were 2–3 years old Korla fragrant pear seedlings with a height of 60–100 cm under conventional water and fertilizer management.

### Different temperatures and cooling agents induced the transition from summer-form to winter-form

For different temperatures treatment, the newly-hatched 1st instar summer-form nymphs (within 12 hr) were transferred to the young vegetative shoots of healthy Korla fragrant pear seedlings at four different temperatures (25 °C, 20 °C, 15 °C, and 10 °C) by using a fine brush. Then, observed the morphological characteristics every 2 days, and counted the number of summer-form and winter-form individuals according to the body color and external characteristics until the 3rd instar. Each treatment was performed with three independent biological replications of at least 30 nymphs for each biological replication.

Menthol (HY-N1369) was purchased from MedChemExpress LLC (Shanghai, China) and borneol (420247) was bought from Sigma-Aldrich LLC (Saint Louis, USA) (*Figure 2—figure supplement 2A-B*; *Peier et al., 2002*; *Turner et al., 2016*). For the treatment of different cooling agents, feeding

setup through a pear leaf was used as previously described in *Aphis citricidus* with minor adjustment (*Shang et al., 2020*). Briefly, a stem with a fresh pear leaf was inserted into a sterile PCR tube of 250 μL containing 200 μL of menthol (1 mg/mL) or borneol (0.1 mg/mL). Then, the PCR tube with leaf was transferred to a 50 mL sterile plastic tube, termed as stem-leaf device. Finally, transferred the newly-hatched 1st instar summer-form nymphs (within 12 hr) into the stem-leaf device, observed the morphological characteristics every two days, and counted the number of summer-form and winter-form individuals according to the body color and external characteristics until the 3rd instar under 25 °C. Each treatment was performed with three independent biological replications of at least 30 nymphs for each biological replication.

## Transmission electron microscopy assay of cuticle

To analyze the difference of nymph cuticle ultrastructure and thickness of thorax between different samples, transmission electron microscopy was used as previously described (*Ekino et al., 2017*; *Ge et al., 2019*). Briefly, the body samples without head were firstly fixed in 4% polyformaldehyde (PFA) for 48 hr and followed by post-fixation in 1% osmium tetroxide for 1.5 hr. Next, the samples were dehydrated in a standard ethanol/acetone series, infiltrated and embedded in spurr resin embedding medium. Then, the superthin sections (–70 nm) of thorax were cut and stained with 5% uranyl acetate followed by Reynolds' lead citrate solution. Finally, the sections were observed, photographed and measured under a Hitachi HT7800 transmission electron microscope operated at 120 kv. Five nymphs were used for each sample.

## Cuticle chitin staining assay with WGA-FITC and determination of cuticle chitin content

Previous studies have demonstrated that cuticle thickness and hardness are important parameters for adapting to environment and natural enemies (*Evans and Sanson, 2005*). Therefore, cuticle chitin staining with WGA-FITC (Cat# F8520, Solarbio, Beijing, China) and cuticle chitin content were measured to analyze the difference between the summer-form and winter-form nymphs, or dsRNA treatment and miRNA agomir treatment. For the WGA-FITC staining, nymph samples were firstly fixed with 4% PFA for 48 hr and dehydrated with a series of sucrose solution with varied concentrations (10% for 4 hr, 20% for 4 hr, 30% for 12 hr), then rinsed three times for 15 min per rinse with sterile PBS buffer. The dehydrated samples were gradually embedded in Tissue-Tek O.C.T. compound (Cat# 4583, SAKURA, Ningbo, China) after the pre-embedding stages under –25 °C. The ultra-thin sections (approximately 70 nm thickness) of embedded material were cut using Leica freezing ultra-cut micro-tome (CM1850, Leica, Weztlar, Germany). The sections were stained with WGA-FITC (50 μg/mL) and DAPI (10 μg/mL) for 15 min and then rinsed three times for 15 min per rinse with sterile PBS buffer as previously described (*Farnesi et al., 2012*; *Xie et al., 2022*). Fluorescence images were acquired using a Leica SP8 confocal microscopy (Weztlar, Germany).

For the determination of cuticle chitin content, chitin Elisa kit (Cat# YS80663B, Yaji Biotechnology, Shanghai, China) was used according to the manufacturer's instruction. In short, each 50 nymphs for one sample were homogenized in 0.2 mL distilled water using an electric homogenizer. The testing samples and standard samples were hatched in enzyme-labeled plates at 37 °C for 30 min, separately. The washing buffer was used to rinse the enzyme-labeled plates five times for 30 s per time. Then, enzyme labeling reagent was added to the plates for hatching and washing. Finally, chromogenic agent A solution and B solution were mixed in holes for coloration at 37 °C and protect from light. The absorbance (OD value) of each hole was measured at 450 nm wavelength and zeroed with a blank hole.

## Sequence characterization of *CcTRPM*

The GenBank accession number of *CcTRPM* was OQ658558. The putative transmembrane domains of CcTRPM were identified by an online software of SMART (a Simple Modular Architecture Research Tool). The tertiary protein structures of *CcTRPM* was predicted with online server Phyre[2] (http://www.sbg.bio.ic.ac.uk/phyre2/html/page.cgi?id=index) and modified with PyMOL-v1.3r1 software. BLASTP in NCBI database was used to search the homologous protein sequences of *CcTRPM* from insects and mammals of different species, including 5 Hemiptera insects, 2 model insects, and 2 mammals. Multiple alignments of the amino acid sequences for *CcTRPM* with other homologies was performed

using DNAman software. Phylogenetic analysis was carried out based on the neighbor-joining (NJ) method using MEGA10.1.8 software.

## qRT-PCR for mRNAs and miRNAs

To examine the temporal expression profile of *CcTRPM*, summer-form samples were collected at different developmental stages, including egg; nymphs of the 1st, 2nd, 3rd, 4th, and 5th instar; adults of 1, 5, and 10 days. For the effect of different temperatures treatment on the mRNA expression of *CcTRPM*, miR-252, and chitin biosynthesis signaling genes, summer-form 1st instar nymphs were treated at 25°C and 10°C. Samples of nymphs were collected at 1, 2, 3, 6, and 10 days after different temperatures treatment and immediately stored at –80 °C. Each sample was performed in three replications, with about 100 individuals for each replication of egg samples and at least 50 insects were included for each nymph or adult samples. All samples were immediately stored at –80 °C for total RNA extraction.

Total RNA was isolated from the above samples using TRNzol Universal (Cat# DP424, TIANGEN, Beijing, China) and miRcute miRNA isolation kit (Cat# DP501, TIANGEN, Beijing, China) for mRNA and miRNA according to the manufacturer' instruction, respectively. The first-strand cDNAs of mRNA and mature miRNA were synthesized from 500 ng of total RNA using PrimeScript RT reagent kit with gDNA Eraser (Cat# RR047A, Takara, Kyoto, Japan) and miRcute Plus miRNA First-Strand cDNA Synthesis Kit (Cat# KR211, TIANGEN, Beijing, China) following the instruction manual. The relative expression of mRNA and miRNA was respectively quantified using TB Green *Premix Ex Taq* II (Tli RNaseH Plus) (Cat# RR820A, Takara, Kyoto, Japan) and miRcute Plus miRNA qPCR Detection Kit (Cat# FP411, TIANGEN, Beijing, China) in a 20 µL reaction mixture on a Bio-Rad CFX Connect Real-Time PCR System (Bio-Rad, Hercules, CA, USA). To reduce the error, two reference genes of *C. chinensis*, *Ccβ-actin* (GenBank accession number: OQ658571) and *CcEF-1* (GenBank accession number: OQ658572), were used as internal controls for real-time PCR. U6 snRNA of *C. chinensis* was used as reference gene for miRNA expression. To check for the specificity, melting curves were analyzed for each data point (*Figure 2—figure supplement 2*, *Figure 4—figure supplement 1*, *Figure 5—figure supplement 1*). The specific reverse primer for miRNA expression was complementary to the poly (T) adapter that was provided in the miRcute plus miRNA qPCR detection kit. qRT-PCR data were quantified using the $2^{-\Delta\Delta CT}$ method, where $\Delta\Delta CT$ is equal to $\Delta CT_{treated\ sample} - \Delta CT_{control}$ (*Livak and Schmittgen, 2001*).

## Ca²⁺ imaging and fluorescence detection of Fluo-4 AM

The ORF sequence of *CcTRPM* was amplified by PCR with specific primers shown in *Supplementary file 1a* and inserted into the pcDNA3.1(+)-mCherry vector to construct the recombinant vector of CcTRPM-pcDNA3.1-mCherry using the *pEASY*-Basic Seamless Cloning and Assembly Kit (Cat# CU201, TransGen, Beijing, China). This recombinant vector was transfected into HEK293T cells using TransIntro EL Transfection Reagent (Cat# FT201, TransGen, Beijing, China) and the cells were cultured for 20–40 hr at 37 °C. These transient-transfected cells were incubated in 500 µL Fluo-4 AM (Cat# S1061, Beyotime, Shanghai, China) for 30 min at 37 °C for staining and washed for three times with Hank's Balanced Salt Solution. Then, these cells were treated with different temperatures, different cooling agents, and used for Ca²⁺ imaging by a Leica SP8 confocal microscopy (Weztlar, Germany) or for fluorescence detection of Fluo-4 AM by a Molecular Devices i3x microplate reader (San Jose, USA).

## Fluorescence in situ hybridization (FISH)

FISH experiment was carried out as previously described with minor adjustment (*Kliot et al., 2014*). Briefly, 20 nymphs for each sample were immersed in Carnoy's fixative (glacial acetic acid: ethanol: chloroform, 1: 3: 6, vol/vol/vol) inside eppendorf tubes for at least 48 hr at room temperature. The fixed samples were washed with 50% ethanol for three times and decolorized in 6% $H_2O_2$ in ethanol for 48 hr at room temperature. Prehybridization was performed with three times for 10 min per time using the hybridization buffer (20 mM Tris-HCl, pH 8.0; 0.9 M NaCl; 0.01% sodium dodecyl sulfate; 30% formamide) without the probe in the dark. The probe sequences of *CcTRPM*-Cy3 and miR-252-FAM are listed in *Supplementary file 1a* and synthesized in GenePharma (Shanghai, China). Hybridize samples for 12 h in the dark in hybridization buffer containing 1 µM fluorescent probe. Combine two

probes with different fluorescent dyes for the co-localization. After washing three times with PBST, cell nuclei were stained with DAPI (1 μg/mL) for 15 min and rinsed with PBST for 15 min. At last, the samples were transferred to the microscope slide, mounted in mounting medium using a cover slip, and viewed under a Leica SP8 confocal microscopy (Weztlar, Germany). The dsCcTRPM treated samples or no-probe samples were used as negative controls for the specificity of probe signal.

## dsRNA synthesis and RNAi assay

Double-stranded RNAs (dsRNAs) of *EGFP*, *CcTRPM*, and 14 transcripts of chitin biosynthesis signaling genes were prepared in vitro using primers ligated with T7 RNA polymerase promoter sequences (*Supplementary file 1a*) using MEGAscript RNAi kit (AM1626, Ambion, California, USA). dsRNAs were further purified with MEGAclear columns (Ambion) and eluted with diethyl pyrocarbonate (DEPC)-treated nuclease-free water. The purity and concentration of dsRNA were then measured via ultraviolet spectrophotometry and gel electrophoresis. The above stem-leaf device was used for dsRNA feeding and RNAi assay.

For the RNAi assays, summer-form 1st instar nymphs were fed with different dsRNAs (500 ng/μL) and divided into three groups. (1) The samples were collected after dsRNAs feeding at 2 days and 4 days under 25 °C condition, and at 3 days and 6 days under 10 °C condition, then for the RNAi efficiency analysis and miR-252 expression by qRT-PCR. (2) Observed the morphological characteristics every two days and counted the number of summer-form and winter-form individuals based on the above description until the 3rd instar under 10 °C condition. (3) The samples were collected after dsRNA feeding at 12 days for cuticle ultrastructure comparison, cuticle chitin staining with WGA-FITC, and determination of cuticle chitin content under 10 °C condition according to the above methods.

For the rescue by menthol feeding under dsCcTRPM treatment, summer-form 1st instar nymphs were fed with the mixture of dsCcTRPM (500 ng/μL) and menthol (1 mg/mL), then counted the number of summer-form and winter-form individuals following the above method until the 3rd instar under 10 °C condition. All the experiments were performed in three replications with at least 30 nymphs for each replication.

## Prediction of miRNAs for *CcTRPM*

The 3'UTR sequence of *CcTRPM* gene was amplified by the 3'-Full RACE Core Set with PrimeScript RTase kit (Cat# 6106, Takara, Kyoto, Japan). miRNAs for *CcTRPM* were predicted by a service provider LC science with two software programs of miRanda (http://www.microrna.org) and Targetscan (https://www.targetscan.org/vert_72/) with the default parameters (*Enright et al., 2003*; *Agarwal et al., 2015*). For software miRanda, the alignment scores of 145 and –15 kcal/mol were set to the threshold. The context score percentile of Targetscan was set as larger than 50 to evaluate the target relationship between miRNA and potential targets. The overlapped miRNAs predicted through both methods were selected.

## In vitro luciferase activity assay

The 419 bp sequence of 3'UTR containing predicted target sites for miR-252 in *CcTRPM* was cloned into the pmirGLO vector (Promega, Wisconsin, USA) downstream of the luciferase gene to construct the recombinant plasmid of CcTRPM-3'UTR-pmirGLO using the *pEASY*-Basic Seamless Cloning and Assembly Kit. The mutated 3'UTR sequence was amplified and cloned into the pmirGLO vector to generate the CcTRPM-3'UTR mutant-pmirGLO plasmid. The agomir and antagomir of miRNAs were chemically synthesized and modified by GenePharma (Shanghai, China) with chemically modified RNA oligos of the same sequence or anti-sense oligonucleotides of the miRNA. The negative control for agomir and antagomir was provided by the manufacturer. HEK293T cells were cultured in a 24-well plate at 37 °C in 5% $CO_2$ overnight and transfected with CcTRPM-3'UTR-pmirGLO plasmid (approximate 500 ng) or CcTRPM-3'UTR mutant-pmirGLO plasmid (approximately 500 ng) and 275 nM agomir-miRNA or agomir-NC using the Calcium Phosphate Cell Transfection Kit (Cat# C0508, Beyotime, Nanjing, China). The activity of the two luciferase enzymes was measured 24 hr after co-transfection following the manufacturer's instruction of Dual-Luciferase Reporter Assay System (Cat# E1910, Promega, Wisconsin, USA). Nine biological replications were carried out for each transfection.

## In vivo RNA-binding protein immunoprecipitation assay (RIP)

A Magna RIP Kit (Cat# 17–704, Merck, Millipore, Germany) was used to perform the RIP assay following the manufacturer's instruction. Specifically, approximately 50 summer-form 1st instar nymphs were fed with agomir-252 or agomir-NC for 24 hr. The treated nymphs were collected and crushed with an auto homogenizer in ice-cold RIP lysis buffer, then stored at –80 °C overnight. Next, 50 µL magnetic beads were incubated with 5 µg of Ago-1 antibody (Merck, Millipore, Germany) or IgG antibody (Merck, Millipore, Germany) to form bead-antibody complex. The frozen lysates were thawed and centrifuged, and the supernatants were divided into three equal parts. One part is considered as an 'input' sample and stored at –80 °C for qRT-PCR analysis. The other two parts were incubated with the magnetic bead-antibody complex (anti-Ago1 or IgG) at 4 °C overnight. The immunoprecipitated RNAs were released by digestion with protease K and extracted by the above methods. Finally, the abundance of *CcTRPM* and miR-252 were quantified by qRT-PCR. The 'input' samples and IgG controls were assayed to normalize the relative expression of target genes and ensure the specificity of the RNA-protein interactions. Six replications were carried out for each treatment.

## Feeding treatment of agomir-252 and antagomir-252

To determine the effect of miR-252 on *CcTRPM* expression, summer-form 1st instar nymphs were fed with agomir-252 (1 µM) or antagomir-252 (1 µM) and collected at 6 days after treatment for qRT-PCR. Agomir/antagomir negative control was fed as the control. To test the effect of miR-252 on the transition of summer-form to winter-form, summer-form 1st instar nymphs were fed with agomir-252 (1 µM) or agomir-NC (1 µM) and divided into three groups. (1) The samples were collected after different agomir feeding at 3 days and 6 days under 10 °C condition, then for miR-252 expression analysis using qRT-PCR. (2) The samples were collected after agomir feeding at 12 days for cuticle ultrastructure comparison, cuticle chitin staining with WGA-FITC, and determination of cuticle chitin content under 10 °C condition according to the above methods. (3) Observed the morphological characteristics every two days and counted the number of summer-form and winter-form individuals based on the above description until the 3rd instar under 10 °C condition.

For the rescue by antagomir-252 feeding under dsCcTRPM treatment, summer-form 1st instar nymphs were fed with the mixture of antagomir-252 (1 µM) and dsCcTRPM (500 ng/µL), then counted the number of summer-form and winter-form individuals based the above method until the 3rd instar under 10 °C condition. All the experiments were performed in nine replications with at least 30 nymphs for each replication.

## Statistical analysis

GraphPad Prism 8.0 software and IBM SPSS Statistics 26.0 were used for making figures and statistical analysis, respectively. All the data were shown as means ± standard deviation (SD) with different independent biological replications. Student's *t*-test was performed for pairwise comparisons to determine the statistically significant differences between treatments and controls (\*\*\*$p < 0.001$). One-way ANOVA followed by the Tukey's HSD multiple comparison test was used for multiple comparisons (different letters denoted by $p < 0.05$).

## Acknowledgements

Thanks for the funding support by the National Natural Science Foundation of China (32202291), China Agriculture Research System (CARS-28), and Chinese Universities Scientific Fund (2023TC048).

## Additional information

### Funding

| Funder | Grant reference number | Author |
| --- | --- | --- |
| National Natural Science Foundation of China | 32202291 | Songdou Zhang |

| Funder | Grant reference number | Author |
|---|---|---|
| China Agricultural Research System | CARS-28 | Xiaoxia Liu |
| Chinese Universities Scientific Fund | 2023TC048 | Songdou Zhang |

The funders had no role in study design, data collection and interpretation, or the decision to submit the work for publication.

## Author contributions

Songdou Zhang, Conceptualization, Resources, Data curation, Formal analysis, Supervision, Funding acquisition, Investigation, Writing – original draft, Project administration, Writing – review and editing; Jianying Li, Resources, Data curation, Formal analysis, Investigation, Methodology; Dongyue Zhang, Zhixian Zhang, Zhen Li, Resources, Software, Formal analysis; Shili Meng, Resources, Software; Xiaoxia Liu, Funding acquisition, Validation, Visualization, Project administration

## Author ORCIDs

Songdou Zhang http://orcid.org/0000-0002-3199-017X
Xiaoxia Liu http://orcid.org/0000-0002-0811-485X

Reviewer #1 (Public Review): https://doi.org/10.7554/eLife.88744.3.sa1
Reviewer #2 (Public Review): https://doi.org/10.7554/eLife.88744.3.sa2
Author Response https://doi.org/10.7554/eLife.88744.3.sa3

# Additional files

## Supplementary files

• Supplementary file 1. The primers used in current study and effect of miR-252 on the transition percent of SF 1st instar nymphs. (a) The primers used in current study. (b) Effect of agomir-252 and antagomir-252 treatment on the transition percent of SF 1st instar nymphs under 10 °C.

• MDAR checklist

## Data availability

The published article includes all data generated or analyzed during this study. The full sequences were submitted to NCBI, accession number: OQ658558 for CcTRPM, OQ734934 for CcTre1, and OQ658570 for CcCHS1.

The following datasets were generated:

| Author(s) | Year | Dataset title | Dataset URL | Database and Identifier |
|---|---|---|---|---|
| Zhang SD, Liu XX, Li JY | 2023 | Cacopsylla chinensis transient receptor potential cation channel subfamily M mRNA, complete cds | https://www.ncbi.nlm.nih.gov/nuccore/OQ658558 | NCBI GenBank, OQ658558 |
| Zhang SD, Liu XX | 2023 | Cacopsylla chinensis trehalase 1 mRNA, complete cds | https://www.ncbi.nlm.nih.gov/nuccore/OQ734934 | NCBI GenBank, OQ734934 |
| Zhang SD, Liu XX, Li JY | 2023 | Cacopsylla chinensis chitin synthase 1 mRNA, complete cds | https://www.ncbi.nlm.nih.gov/nuccore/OQ658570 | NCBI GenBank, OQ658570 |

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
