## [Editor Report · eLife assessment]

This is a **valuable** study of the molecular basis of summer-to-winter transition in the pear psyllid pest, *Cacopsylla chinensis* (hemiptera). The molecular and organismal experiments using current methodologies to evaluate the cold responsiveness of the target proteins are mostly **convincing**, but the structural and phylogenetic analyses remain inconclusive. The results of this study will be of interest to entomologists.

---

## [Referee Report · Reviewer #1 (Public Review)]

Here, in this revised manuscript, the authors describe the transition between the summer form and the winter form of the pear psyllid pest, Cacopsylla chinensis (hemiptera). While the authors explore many components of this transition, the central hypotheses they seek to test are (i) that a protein they deem CcTRPM is a cold-sensitive Transient Receptor Potential Melastatin (TRPM) channel, and (ii) that this channel is involved in the summer-to-winter transition, in response to cold.

The authors demonstrate that: both cold and menthol can initiate the summer-to-winter transition; that the protein of interest is required for the summer-to-winter transition (in vivo); that the protein of interest is involved in menthol- and cold-dependent Ca2+ transients (in vitro); that miR-252 expression is temperature-dependent, modulates the seasonal transition, and affects the expression of the transcript of interest; and finally, somewhat separately, that the chitin biosynthesis pathway is linked to the summer-to-winter transition.

However, I note three weaknesses, which are largely inherited from the original manuscript.

Firstly, the identification of the TRPM gene seems to be partially couched in the ab initio structural identification of "conserved ankyrin repeats." The methodology used to identify these so-called ankyrin repeats is not sufficiently described, and their conserved status is not sufficiently demonstrated nor cited (to my knowledge, this would be the first description of ankyrin repeats in TRPM, whereas previous studies have not detected them). There is also no discussion of previously identified structural components of TRPMs (see: Yin et al 2018, DOI: 10.1126/science.aan4325)

Secondly, the phylogenetic analysis still appears to be incomplete. The authors claim that "insects TRPM and mammals TRPM belong to different branches in evolution." While this is not a paper centered on the evolutionary analysis of this gene/protein family, the phylogenetic analysis here is insufficient for justifying this claim, especially since this claim is counter to previous studies (many in the literature over the past 10 years).

Thirdly, the methods lack sufficient detail to completely reproduce the phylogenetics and the cold-induced Ca2+ imaging.

Despite these weaknesses, I find the organismal/molecular component of this manuscript to be clear and convincing.

---

## [Referee Report · Reviewer #2 (Public Review)]

The pear psylla Cacopsylla chinensis has two morphologically different forms, winter- and summer-forms depending on the temperatures. The authors provided solid data showing that the cold sensor CcTRPM is responsible for switching summer- to winter forms, which is in turn regulated by the miRNA miR-252. This finding is interesting and novel.

---

## [Author Response]

The following is the authors’ response to the original reviews.

Thanks for your comments and suggestions concerning our manuscript entitled “miR-252 targeting temperature receptor CcTRPM to mediate the transition from summer-form to winter-form of Cacopsylla chinensis”. These comments are all of great important and extremely helpful for revising and improving our manuscript. We have revised the manuscript carefully according to all your comments. Our point-by-point responses to the comments are listed below.

**Reviewer #1 (Recommendations For The Authors):**
1. If the authors wish to improve their phylogenetic analysis, I strongly suggest using their hemipteran sequences alongside the *Drosophila* homolog and at least all of the human paralogs. This should be generally sufficient to recapitulate the generally accepted TRPM phylogeny. If the authors contend that this is in fact a separate lineage from other insect TRPMs, a phylogeny that is as taxonomically inclusive as possible, and as methodologically rigorous as possible, would be ideal.

Thanks for your great suggestion. We have redid the phylogenetic analysis in Figure S1B using CcTRPM sequence with homologs from other 16 species, including 8 human paralogs, 1 *Mus musculus* homolog, 1 *Drosophila* homolog, and 6 insect homologs. The relative description was added in Line 489-491 and Line 1044-1049 of our revised manuscript.

1. If the authors wish to conclude that this is a cold-sensitive ion channel, I strongly suggest repeating at least the Ca2+ imaging with a cold stimulus. In the absence of this experiment, I think that the conclusions need to be significantly softened/hedged, making it clear that the only evidence of cold sensitivity is indirect (resulting from the knockdown experiments).

Thanks for your excellent suggestion. We have performed Ca2+ imaging with a cold stimulus of 10°C. As expected, there was a clear increase of Ca2+ concentration was observed when treated with cold stimulus of 10°C, which was similar with menthol treatment. So, we could get the solid conclusion that CcTRPM is a direct cold-sensitive ion channel in C. chinensis. We also have added the Ca2+ imaging result with a cold stimulus of 10°C in Figure 2D and moved the results of Ca2+ imaging with menthol treatment to Figure S2I. The related results and methods were added in Line 193-200, Line 919-923, and Line 1065-1069 of our revised manuscript.

1. Lines 173 and 181: The method used to identify the putative transmembrane domains was not described (although the 3D model does have the correct TRP structure, these methodological details would be appreciated).

Thanks for your great suggestion. We used an online software of SMART (a Simple Modular Architecture Research Tool) to identify the putative transmembrane domains of CcTRPM, and have added these methodological details in Line 485-487 of Materials and Methods of our revised manuscript.

1. Lines 176-178: The authors state that "phylogenetic analysis revealed that CcTRPM was most closely related to the DcTRPM homologue (Diaphorina citri, XP_017299512.2), which was consistent with the evolutionary relationships predicted from the multiple alignment of amino acid sequences." The meaning of this sentence is unclear to me. I'm not sure what it means to be "consistent with the evolutionary relationships predicted from the multiple alignment of amino acid sequences."

Thanks for your excellent suggestion. We have revised this sentence in Line176 to 179 of our revised manuscript.

1. Lines 474-475: The authors state that the NCBI database was used to identify homologous sequences, but there isn't sufficient methodological detail to repeat the search. For example, was this a BLASTP search? Was it taxonomically restricted? What statistical thresholds for homology inference were used? These details would be much appreciated.

Thanks for your great suggestion. We used BLASTP of NCBI database to identify homologous sequences and preferred the representative species that TRPM sequences have been reported. We have added more description about the methodological detail of phylogenetic analysis in Line 489 to 491 of our revised manuscript.

1. It would be very interesting, but not critical, to know if menthol and borneol alone have an effect on cuticle thickness.

Thanks for your excellent suggestion. Actually, we performed the experiments of menthol and borneol alone on cuticle thickness at the beginning. Under 25°C condition, treatment of menthol and borneol alone induced 30-40% transition of 1st instar nymphs from summer-form to winter-form, but only had some slight effect on cuticle thickness, not strong as 10°C of low temperature, because of the opposite effect of 25°C. However, under 10°C condition, we could not know whether the effect on cuticle thickness is from 10°C of low temperature, or direct from menthol and borneol alone.

1. It would be interesting, but not critical, to confirm the authors' ab initio protein folding by comparing their model to the AlphaFold2-derived model, either by folding it themselves or extracting it from the AlphaFold Protein Structure Database, if it has already been folded by DeepMind.

Thanks for your great suggestion. We have predicted the tertiary protein structures of CcTRPM with AlphaFold2 software and the result was shown in Author response image 1. Compared with the result in Figure 2A, the conserved ankyrin repeats (ANK) and six transmembrane domains were almost similar.

**Author response image 1. sa3fig1:** The tertiary structures of CcTRPM predicted with AlphaFold2 software.

1. Figures 1F-G, 3F, 4A-B, 5G-J, S6C, and S7C-D do not plot replicates (although these are plotted in other figures).

Thanks for your excellent suggestion. Besides Figure 1F-G was stacked grouped graph type and could not add the plot replicates, we have added the plot replicates in Figures 3F, 4A-B, 5G-J, S6C, and S7C-D of our revised manuscript.

1. Figure 5A-C, and associated text: The significance of these findings is somewhat lost on me, coming from a position of general naivety concerning chitin biosynthesis. My interpretation of Figure 5A was that each of these steps was a necessary component of chitin biosynthesis. It was thus surprising that not all of the steps were required. I think it would be exceptionally helpful if the authors spent more time describing this pathway, alternative pathways to generating the intermediate steps, and ultimately, their hypothesis of why only two steps seem critical.

Thanks for your great suggestion. The signal pathway of chitin biosynthesis in Figure 5A was modified from the paper of Doucet and Retnakaran, 2012. De novo biosynthesis of chitin has eight enzymatic steps, including 1 Trehalose, 2 enzymes in Glycolysis, 4 enzymes in Hexosamine pathway, and 1 Chitin synthesis. Glycolysis and hexosamine pathway are two complex cellular metabolic processes within organisms. We supposed that there are two reasons for not all of these steps were required: (1) the function of some enzymes may be replaced or supplemented by other enzymes, for examples, function of hexokinase and glucokinase was similar. (2) The reason for no obviously phenotypic defects might be cause by insufficient interference efficiency of RNAi. So, it’s worth to further study the functions of these chitin biosynthesis enzymes by CRISPR-Cas9 in future. We have added more describing about this chitin biosynthesis pathway in Line 379-390 of our revised manuscript.

**Reviewer #2 (Recommendations For The Authors):**
1. Line 19, should be morphological transition.

Thanks for your excellent suggestion. We have changed “behavioral transition” to “morphological transition” in Line 19 of our revised manuscript.

1. Line 21, delete the novel.

Thanks for your excellent suggestion. We have deleted the word of “novel” in Line 21 of our revised manuscript.

1. Fig. 2B, did authors examine the CcTRPM expression level before 3 d? Given that CcTRPM acts as a cold sensor, it is supposed to respond to temperature change quickly.

Thanks for your excellent suggestion. We have examined the CcTRPM expression level in 1 d and 2 d after 10°C treatment compared with 25°C treatment. As expected, CcTRPM expression levels were also obviously increased in 1 d and 2 d after 10°C treatment. We have added the relative results in Figure S2F and relative description in Line 184-185, Line 500, and Line 1059-1060 of our revised manuscript.

1. Fig. 2I, from the figure legend and the text in the panel, it's hard for readers to understand what the authors intend to say. This data is important since knockdown of CcTRPM decreases the winter-form from 90% to 30% at 10℃. Provide more information in the figure legend.

Thanks for your excellent suggestion. We have added more information in the figure legend of Figure 2I in Line 933-939 of our revised manuscript.

1. Line 224, ...CcTRPM functions as a molecular switch to modulate the transition from .... The phrase 'molecular switch' is inappropriate because knockdown of CcTRPM partially decreases the form ratio as shown in Fig.2I instead of reversing the effect completely. So, use other words instead of 'molecular switch'.

Thanks for your excellent suggestion. We have changed “a molecular switch” to “an essential molecular signal” in Line 225 of our revised manuscript.

1. Fig. 4G, this data is important. It's nice to see that this data is provided.

Thanks for your excellent suggestion. We have provided the data of Figure 4G in Table S2 of our revised manuscript.

1. Authors showed that CcTRPM functions as a cold receptor to regulate the transition of C. chinensis from summer-form to winter-form. Does this mean that a heat receptor gene functions oppositely by transiting winter-form into summer-form? Did the authors test the function of a heat TRP in the form transition? At least, discuss this in the discussion part.

Thanks for your excellent suggestion. TRPV ion channel has been reported to function as a heat receptor in mammals by David Julius (Caterina et al., 1997; Cao et al., 2013). So, we supposed TRPV maybe function as a heat receptor to induce the transition from winter-form to summer-form in C. chinensis. The relative tests are on going. We have added two references in Line 681-686 and some discussion about the heat receptor in Line 341-345 of our revised manuscript.

1. Line 433, which tissue was used for transmission electron microscopy?

Thanks for your excellent suggestion. The thorax was used for transmission electron microscopy, and we have added the information in Line 448 and Line 453 of our revised manuscript.

1. How is the conservation of miR-252? Does the regulatory role of CcTRPM and miR-252 apply to the psylla family in addition to C. chinensis?

Thanks for your excellent suggestion. Besides C. chinensis, the phenomenon of summer-form and winter-form also existed in other psylla species, like Cyamophila willieti. Because of no genomic information was reported in most psylla species, we could not evaluate the conservation of miR-252 between different psylla species. However, it is worth and interesting to clarify whether the function of TRPM and miR-252 were conserved in the future.